# OpenReview forum: "Dynamic Relational Priming Improves Transformer in Multivariate Time Series"
_ICLR.cc/2026/Conference — Submitted to ICLR 2026_

### Official Review · Reviewer_yrUp · 2025-10-29

**Soundness:** 2
**Presentation:** 3
**Contribution:** 2
**Rating:** 4
**Confidence:** 4

**Summary:**

This paper proposes Prime Attention, a modified attention mechanism for multivariate time series (MTS) forecasting. Standard attention uses fixed token representations for all pairwise interactions, termed static relational learning by the authors. Prime Attention introduces dynamic relational priming through learnable, pair-specific filters. The authors formalize this distinction between static and dynamic approaches in a theoretical framework.

**Strengths:**

- The conceptual distinction between static and dynamic relational learning is clear and useful, providing good motivation for the method.

- The method shows consistent but modest improvements on heterogeneous MTS datasets like Weather and Solar-Energy.

- The module can replace standard attention in multiple backbone architectures, demonstrating modularity and broad applicability.

**Weaknesses:**

- Scalability is a major concern. The $O(N^2 \times d)$ memory overhead for pairwise filters ($F_{i,j}$) makes the method impractical for high-dimensional MTS. Claims about maintaining asymptotic complexity are misleading.

- The paper mentions a GNN-based sparsification strategy in the appendix to address scalability  but provides no empirical evaluation. Its viability and performance trade-offs remain unknown.

- Implementation details for the filters ($F_{i,j}$) are unclear. How are filters parameterized24? Was the FFT-based initialization actually used in experiments25252525?

**Questions:**

- While authors argue its improvement, but results lack statistical rigor. Are the reported performance improvements statistically significant? Please report results averaged over multiple random seeds with variance or confidence intervals.

- Can you provide a quantitative evaluation of the GNN-based sparsification strategy from Appendix? What is the resulting performance versus memory trade-off?

- Was the FFT-based lead-lag initialization (Equation 13) used in the main experiments? If so, for which datasets, and what is its impact compared to random initialization?

- How are the $F_{i,j}$ filters parameterized for high-dimensional datasets? Is the generating MLP shared across all pairs, or are parameters learned for each pair independently?

- Why does the method fail to produce significant gains on homogeneous datasets like ECL and Traffic? What properties of these datasets limit the method's effectiveness?

---

> ### Author Response · Authors · 2025-11-17
> **Author response (A1, A2)**
>
> Thank you for your thorough review. We appreciate your insightful comments, and we aim to improve our paper by addressing your questions and concerns.
> ### A1 on statistical significance of results
> Thank you for pointing this out. We use averaged results over 5 runs with random seeds. We find stable performance across the board with negligible variance for the most part. We will make our experimental details more clear in the appendix. We provide a snippet of the full results (in MSE) below (using iTransformer as backbone and using random seed in each run for 5 runs).
>
> | Horizon \ Dataset | ETTh1             | ETTh2            | Weather          | ECL              | Traffic          |
> | ----------------- | ----------------- | ---------------- | ---------------- | ---------------- | ---------------- |
> | 96                | $0.378\pm 0.0005$ | $0.297\pm0.001$  | $0.170\pm0.0002$ | $0.146\pm0.0005$ | $0.395\pm0.0001$ |
> | 192               | $0.432\pm0.0006$  | $0.376\pm0.001$  | $0.218\pm0.0002$ | $0.164\pm0.001$  | $0.417\pm0.0002$ |
> | 336               | $0.473\pm0.0009$  | $0.413\pm0.001$  | $0.275\pm0.0002$ | $0.179\pm0.001$  | $0.431\pm0.0001$ |
> | 720               | $0.479\pm0.001$   | $0.420\pm0.0008$ | $0.354\pm0.0005$ | $0.214\pm0.002$  | $0.463\pm0.001$  |
> | Avg.              | $0.441\pm0.0007$  | $0.377\pm0.001$  | $0.254\pm0.0002$ | $0.176\pm0.001$  | $0.427\pm0.0003$ |
>
> ### A2 GNN-based sparsification trade-off and memory usage
> We find that performance generally increases with higher memory when using the GNN-based sparsification strategy. When compared against standard attention (as a fully connected graph), GNN-based prime attention can achieve comparable performance using up to 40% less connectivity (section G.2). This results in lower "effective" complexity compared to standard attention, **ultimately using less memory to achieve better performance than standard attention**. This is related, but separate, from the sequence length analysis done in section G.4, which shows that prime attention achieves better performance than standard attention using less input data (this concerns actual sequence length per channel, not channel connectivity), which also results in lower "effective" memory usage in prime attention. We provide the performance (in MSE) vs. memory usage on the Weather dataset below (we provide this as a table here for latex compatibility but will make it a chart in the updated paper for better visualization).
>
> | Adj. Density   | Memory     | Performance |
> | -------------- | ---------- | ----------- |
> | 0.2            | 765.21 MB  | 0.180       |
> | 0.4            | 1212.75 MB | 0.179       |
> | 0.6            | 1691.39 MB | 0.177       |
> | 0.8            | 2177.59 MB | 0.176       |
> | 1.0 (full adj) | 2664.64 MB | 0.175       |
> For reference, standard fully-connected attention achieves 0.177 MSE using 2102.92 MB in this setup, requiring around 400 MB MORE than prime attention for comparable performance.
>
> **Response to scalability as a major concern**: We agree with the reviewer that model scalability is a major concern. With section G.2 and G.4 both showing, independently, that prime attention can outperform standard attention with lower memory usage by using less connectivity and less sequence length, we believe this provides a strong justification for prime attention's practical adoption.
>
> As for existing benchmarks, we were able to use prime attention without any compromise even on high-dimensional datasets like Traffic (N=862), further demonstrating prime attention's utility. As for situations with truly massive channel-count (1000+), even standard attention would likely begin facing context-size issues in deployed systems and would likely resort to sampling strategies anyway (which can be seen as an instance of controlling connectivity density, and as shown in section G.2, prime attention is favorable in such scenarios).
>
> As for our characterization of "maintaining asymptotic complexity" being misleading, we fully acknowledge that prime attention introduces sizeable added computation but maintain that our statement is both factual and consistent with scaling laws for element-wise computation as introduced by prime attention. We also make clear in the paper that prime attention approximately doubles the FLOP count and therefore believe that we were engaging in good-faith and transparency. If the reviewer still finds some statements misleading, we are happy to discuss this further and make adjustments where necessary.

---

> ### Author Response · Authors · 2025-11-17
> **Author response (A3-A5)**
>
> We answer the rest of the reviewer questions and concerns here.
>
> ### A3 on filter initialization strategies
> The FFT-based initialization was used on all datasets for consistency. We find that FFT-based initialization is generally better than random initialization but this is not always the case. For example, on the exchange dataset, random initialization outperforms FFT-based initialization by around 0.6\%, perhaps because predictable patterns like lead-lag are not as apparent. On datasets like weather with known patterns, FFT-based initialization outperforms random initialization by around 1.5\%. Overall, we find that both strategies are capable of improving prime attention's performance and we would recommend using FFT-based initialization on datasets with known patterns and random initialization when prior knowledge either does not exist or is unreliable. We will update the appendix to make this more clear.
> ### A4 on filter parameterization strategy
> The parameters to generate $\mathcal{F}$ filters are SHARED across all pairs, including high-dimensional datasets. We find that this strategy works well and introduces parameter count increase of around 1-2\% compared to standard transformers. This also makes it easier to perform fair comparison with standard transformers since the parameter count increase on prime attention is negligible. We will update the appendix to make this clear.
> ### A5 on performance on homogeneous datasets
> Rather than prime attention being ineffective on homogenous datasets, we believe it is more about standard attention being good enough to capture most patterns in these datasets. As such, we believe in these datasets, prime attention simply does not have much room for improvement beyond marginal gains. This is consistent with our inductive bias hypothesis and strengthens our claims that prime attention is designed for heterogenous information mixing while standard attention should still be capable of showing great performance on homogenous data.

---

> ### Comment · Reviewer_yrUp · 2025-11-26
>
> The clarification about the proposed method and shared filter parameters resolves the ambiguity. In addition to that, the new table for the Weather dataset quantifies the memory and performance trade off for the sparsification setting.
>
> However, the efficiency analysis uses only the Weather dataset. No empirical evidence shows that the reported memory trade off extends to high dimensional settings such as Traffic where sparsity is important. The continued use of the dense ($O(N^{2})$) implementation for the main benchmarks leaves the practical limit of the method untested. Furthermore, the drop in accuracy on the Exchange dataset under FFT initialization suggests that the inductive bias depends on the dataset. Could you elaborate more  details regarding these concerns?

---

> > ### Author Response · Authors · 2025-11-28
> > **Author response to reviewer's comment**
> >
> > Thank you for your response to our rebuttal. We appreciate the reviewer's thorough audit of both our paper and our rebuttal and aim to address all additional and remaining concerns.
> > ### On Memory/Performance Tradeoff and Practical Limits of Prime Attention
> > On higher dimensional datasets such as Traffic and ECL, we found that the memory and performance tradeoffs were actually much more pronounced in favor of prime attention. In fact, at just 20\% connectivity, prime attention consistently outperforms standard attention operating at 100\% connectivity (full attention), using 67\% less memory as a result. Unlike Weather however, both prime and standard attention actually exhibited their best performance at low connectivity (around 20\%) and their performance did not show improvements with higher connectivity (with prime attention still consistently outperforming standard attention). In general, we hypothesize that homogenous datasets with largely homogenous inter-channel patterns such as Traffic and ECL may actually benefit from sparsity as a form of regularizer (given that self-attention is always enabled) and that 100\% connectivity may actually be hindering model performance due to undue noise accumulation from large receptive field. However, we decided to steer clear of these results as we believe sampling dynamics are outside the scope of this work.
> >
> > As the reviewer points out, the practical limits of prime attention has not been tested. Because the memory/performance tradeoff analysis was not clean-cut on all datasets likely due to dataset characteristics and sampling dynamics, we provide a more controlled test on prime attention's practical limits below.
> >
> > | channel count | memory (MB) |
> > | ------------- | ----------- |
> > | 200           | 910.97      |
> > | 400           | 3533.47     |
> > | 600           | 7887.04     |
> > | 800           | 13990.02    |
> > | 1000          | 21836.74    |
> > | 1200          | 31433.40    |
> > | 1400          | 42761.18    |
> > | 1600          | 55836.52    |
> > | 1800          | OOM         |
> > | 2000          | OOM         |
> > We create a synthetic dataset with variable channel-count filled with dummy data and record peak memory usage during training. The model is run on a single NVIDIA A100 80GB GPU using the same experimental settings as the Traffic dataset. We observe quadratic memory growth and OOM starting with receptive field of 1800 channels. We will include this as a chart in our ablations.
> >
> > While we certainly confirm that memory growth is an issue, we believe that being able to run nearly twice the channel-count of the largest benchmark dataset (Traffic) shows practical utility. Overall, we would argue that the introduction of prime attention as a technique itself is the main contribution of our paper. While we start the conversation with our GNN-based sparsification strategy, we leave prime attention's adoption to truly massive datasets (with N > 1800) to future work.
> > ### On Variability of Inductive Bias on Different Datasets
> > We agree with the reviewer's sharp analysis of the inductive bias varying on the dataset. While all initialization strategies we tested for prime attention still outperform standard attention, we do notice that there is still room for optimization. In general, we believe that explicit initialization should be used when known patterns exist while random initialization (and therefore learned from scratch) are a good out-of-the-box solution for datasets with unknown or unpredictable patterns.

---

### Official Review · Reviewer_arqk · 2025-10-31

**Soundness:** 3
**Presentation:** 2
**Contribution:** 2
**Rating:** 4
**Confidence:** 4

**Summary:**

This paper aims to address the "domain mismatch" problem when applying Transformers to MTS forecasting. The authors argue that standard attention's "static relational learning" (i.e., fixed K/V representations) fails to capture the inherent "heterogeneous" dependencies in MTS data. To this end, the paper proposes "dynamic relational priming attention" (prime attention), the core of which is to introduce a learnable, pair-specific $(i, j)$ "primer" $\mathcal{F}_{i,j}$. This primer dynamically modulates the K/V vectors via element-wise multiplication ($\odot$) *before* the attention calculation. Experiments show that this mechanism improves performance when replacing SOTA models, especially on datasets with high heterogeneity.

**Strengths:**

* The paper's diagnosis of the conflict between "static K/V representations" and the "heterogeneity" of MTS data is both sharp and reasonable. Linking the limitations of the attention mechanism to the specific properties of the domain (heterogeneity) provides a solid foundation for the paper.

* A key strength of this paper is that its experimental design is tightly coupled with its core hypothesis. The results strongly support this hypothesis: the method shows significant improvements on datasets considered "heterogeneous" (like Weather and Solar) but "marginal" improvements on those considered "homogeneous" (like ECL and Traffic). This validates that the method is indeed solving the specific problem it claims to address.

* **Insightful Efficiency and Mechanism Analysis**:
    * The sequence length efficiency analysis (Fig. 1) is a highlight, suggesting that Prime Attention, as an effective inductive bias, may allow the model to learn relationships more efficiently.
    * The attention map analysis in Appendix G.3 (Fig. 6, 7) is particularly convincing. It demonstrates the model's ability to adapt based on data characteristics: on the Solar dataset, it learns a "channel-independent"-like behavior, while on the Exchange dataset, it learns to strengthen cross-channel interactions. This powerfully demonstrates the "dynamic" and adaptive nature of the mechanism.

**Weaknesses:**

* **Over-reliance on the Appendix for Structural Integrity**:
    This is a notable structural issue. The main paper is not fully self-contained. Some of the most critical evidence supporting the core arguments is placed entirely in the appendix. For example, the **attention map analysis (Appendix G.3)**—which should be a central figure in the main body—as well as key ablation studies (G.1) and empirical complexity analysis (G.2), are not in the main text. This makes it difficult for a reader to coherently evaluate the paper's core claims within the main body and weakens its persuasiveness.

* **Insufficient Justification for the Novelty of the Core Mechanism**:
    1.  **Gating Mechanism**: The core mechanism ($\odot \mathcal{F}_{i,j}$) is a form of gating. However, the paper fails to sufficiently argue why this specific pair-wise gating (which occurs *inside* the attention calculation and carries an $\mathcal{O}(N^2)$ cost) is a superior design choice compared to other "global" gating strategies in the field (e.g., fusing "interactive" and "non-interactive" representations *outside* the attention calculation).
    2.  **Comparison with MHA**: The paper categorizes standard MHA as "static" in Section 4. This argument may be an oversimplification, as it overlooks that MHA itself is a mechanism for capturing multiple, parallel dynamic patterns. The paper **lacks a direct experimental comparison** to demonstrate that its explicit $\mathcal{F}_{i,j}$ parameterization (with $\mathcal{O}(N^2)$ cost) has a fundamental advantage over a standard MHA with an equivalent parameter count (and its *implicit* modeling capabilities).

**Questions:**

1. **Pair-Specific vs. Global Gating**: Can you provide experimental or theoretical justification for why your pair-specific gating mechanism $\mathcal{F}_{i,j}$ outperforms global gating strategies, such as those in [1]?

2. **Comparison with Multi-Head Attention**: Could you conduct a fair comparison between Prime Attention and standard Multi-Head Attention (MHA) with an equivalent total parameter count (e.g., by adjusting $d_{model}$ or the number of heads $h$) to demonstrate the necessity of $\mathcal{F}_{i,j}$?

3. **Marginal Improvement on Homogeneous Datasets**: Given the marginal improvement on homogeneous datasets like ECL, do the learned $\mathcal{F}_{i,j}$ primers approach an identity vector, effectively making Prime Attention degenerate into standard attention? An analysis of the $\mathcal{F}$ matrix in this case would be valuable.

**Details Of Ethics Concerns:**

1.  As mentioned in the "Weaknesses," your $\mathcal{F}_{i,j}$ is a pair-specific gating mechanism. Could you provide an experimental comparison or a deeper theoretical analysis to explain why you believe this (intra-attention) gating strategy is superior to other "global" gating strategies[1]?

2. To more clearly justify the necessity of your proposed mechanism, could you provide a "fair" comparison experiment against standard Multi-Head Attention (MHA)? Specifically, a comparison of Prime Attention against a standard MHA with an **equivalent total parameter count** (e.g., by increasing $d_{model}$ or the number of heads $h$). This would be key evidence to prove the necessity of your explicit $\mathcal{F}_{i,j}$ parameterization.

3. Your finding that the model's improvement is "marginal" on "homogeneous" datasets like ECL is very interesting. We are curious about what the model learns in this scenario: do the learned $\mathcal{F}_{i,j}$ primers converge to something close to an identity vector ($\mathbf{1}$), thereby causing Prime Attention to automatically degenerate into standard attention? An analysis of the $\mathcal{F}$ matrix in this context would help us understand the mechanism's adaptive behavior when heterogeneous signals are absent.

[1] Lan, Yu-Hsiang, and Eric K. Oermann. "Gateformer: Advancing Multivariate Time Series Forecasting through Temporal and Variate-Wise Attention with Gated Representations." arXiv preprint arXiv:2505.00307 (2025).

---

> ### Author Response · Authors · 2025-11-13
> **Response to global vs pair-wise gating questions**
>
> Thank you so much for your valuable feedback. We hope in answering your insightful questions and concerns, we were able to clarify and even improve our work.
>
> ### For Q1 on pair-specific vs global gating:
> Thank you for pointing out this perspective of viewing prime attention as a form of pair-specific gating. We largely agree with this perspective but argue that global gating used in works like GateFormer [1] is fundamentally different not only in where they are applied (pair-specific vs. global) but also in what they end up achieving. As such, we believe it is not straightforward to decide which method is "superior". Rather, we think these are adjacent directions solving different but equally useful problems in MTS. GateFormer uses gating to fuse information within each token to strengthen token representations, not make them more well-aligned for pattern discovery. On the other hand, prime attention is a way to modulate token representations based on pair-wise dynamics in order to improve pattern discovery between different channel pairs and their specific pair-wise context.
>
> **Theoretically, GateFormer remains a static relational learning method** under Definition 4.1 and Theorem 4.3. One simply needs to replace $h_{i \rightarrow j} = x_iW_q$ in theorem 4.3 with $s^{(i)} = a \odot v^{(i)}_T  + (1-a)\odot v^{(i)}_G$ from GateFormer, where both $v^{(i)}_T$ and $v^{(i)}_G$ are from $x_i$, representing intra-token fusion/gating. **Therefore, Gating used in GateFormer is more of a facilitation of intra-token fusion to learn an effective representation, rather than inter-token, pair-wise modulation used in prime attention.**
>
> In addition, our gating also differs mechanistically from GateFormer as prime attention is able to amplify information unlike GateFormer, which uses $a$ and $(1-a)$ for $0 \le a \le 1$. In this sense, **we would posit that prime attention's mechanism is closer to adaptive filtering rather than a strict "gating" operation**. This does not mean that either work is superior to each other in any way, though we believe prime attention may still be more robust in highly heterogenous datasets. Prime attention is on par with GateFormer's reported performance though **direct comparisons are difficult because prime attention only uses linear layers for tokenization (to keep things consistent with iTransformer and others) while GateFormer employs more sophisticated (non-linear) techniques as core part of its tokenization process**.
>
> Overall, we believe that prime attention and GateFormer are works tackling unique issues at different stages of the modeling process, and are therefore complementary works to be used in conjunction, rather than competing works to compare against.

---

> ### Author Response · Authors · 2025-11-13
> **Author response to rest of the review**
>
> ### For Q2 on fair comparison between prime and std MHA:
> Here, we conduct a fair comparison between standard MHA and prime attention by increasing the number of heads (along with overall model dimensions to keep per-head dimensions consistent) in standard MHA. Results in MSE are shown in the table below and lower values indicate better performance. Because the learned primers are calculated once before each transformer block (they are used in each attention layer but calculated once) and due to shared parameters in the MLP during generation of $\mathcal{F}$, the **parameter increase in prime attention is actually marginal**, **and accounts for only 1-2% increase in parameter count**. Due to this, it was hard to match standard MHA's parameters exactly with prime attention since adding even 1 additional head in standard MHA over-shoots prime attention's parameter count increase by around 35%. Despite this, prime attention still outperforms MHA with 35% more parameter count than prime attention while MHA shows very marginal improvement (or slight degradation) with increased parameter count. **This shows that performance hits a wall with standard MHA, where simply increasing parameter count does not allow the model to learn better patterns. We believe this strengthens our work, displaying the diminishing returns of model size in standard MHA while showing that prime attention can get past this with marginal increase in parameter count.**
>
> | Data\ Model | std. MHA | std. MHA with increased params | prime attention |
> | ----------- | -------- | ------------------------------ | --------------- |
> | ETTh1       | 0.386    | 0.385                          | 0.378           |
> | Weather     | 0.176    | 0.176                          | 0.170           |
> | Solar       | 0.207    | 0.209                          | 0.199           |
> On the point that MHA itself is a mechanism for capturing multiple, parallel, dynamic patterns, we agree with the reviewer. We believe this comes down to the value of inductive bias given dataset complexity as well as how well standard attention can actually learn these diverse pair-wise patterns in practice. For datasets like the ETT variants for example, even though there are only 7 channels and standard MHA could theoretically learn every pair-wise relationship with 7+ total heads, prime attention outperforms standard MHA. While this does not mean that bigger channel count necessarily favors prime attention, it does indicate that learnable inductive bias of prime attention facilitates better pattern extraction and more meaningful learning even when standard MHA is technically capable of comparable performance.
> ### For Q3 on filter analysis on homogenous datasets
> This is an extremely insightful suggestion and we appreciate you suggesting filter value analysis to see what is happening when working with homogenous datasets. Interestingly, we do NOT observe filter values collapsing to the identity on these datasets and thus they are not learning to revert back to standard attention. Instead, we find that most values are zero-centered, with very few values being amplified with max magnitude of around 1.5. We observe similar trends in heterogenous datasets as well. This suggests that prime attention is learning a different pattern, one that suppresses most dimensions while making a few dimensions dominate the representation. We believe this may be in response to the sparsity of informative signals in multivariate time series (even on homogenous datasets). Furthermore, **this helps explain why channel-independent (CI) models can be quite competitive against channel-mixing models by showing that most inter-channel data are non-informative**. In light of this, **we believe this actually strengthens the case for prime attention, achieving the best of both worlds by learning from inter-channel information in the rare cases where they are useful while learning to cancel out the noise from non-informative inter-channel data in the majority of cases**.
> ### On over-reliance on appendix
> We agree with the reviewer and will move critical analyses (such as key ablation and attention map analysis) to the main text. We believe this can be a straightforward procedure, especially given the extra page allowance for final version.

---

### Official Review · Reviewer_1z1E · 2025-11-01

**Soundness:** 3
**Presentation:** 3
**Contribution:** 3
**Rating:** 6
**Confidence:** 3

**Summary:**

This paper identifies a fundamental mismatch between the static relational learning of standard attention mechanisms and the heterogeneous nature of multivariate time series (MTS) data. The authors argue that in domains like NLP, token relationships are relatively homogeneous (e.g., semantic), whereas in MTS, different pairs of channels (e.g., temperature-pressure vs. temperature-flow) can be governed by entirely different physical laws and temporal dynamics (e.g., lead-lag vs. instantaneous correlation). Standard attention, which uses a single, fixed token representation (K, V) for all pairwise interactions, is ill-suited for this heterogeneity.
To address this, the paper introduces a novel theoretical framework to formalize this problem, defining "Static Relational Learning" (where a token's representation is independent of its interaction partner) and "Dynamic Relational Learning" (where a token's representation is conditioned on its partner). The paper provides strong empirical validation, showing that simply swapping standard attention for prime attention in SOTA Transformer backbones (iTransformer, Timer-XL, FreDF) yields consistent and significant forecasting improvements (up to 6.5%).

**Strengths:**

S1. This paper is well-motivated and solves the timely and important problem exist in the time-series forecasting domain

S2. The paper is well-written and easy to follow

S3. The experimental results show that proposed methods significantly improved the performance, which empirically validate their hypothesis.

**Weaknesses:**

W1. The most significant practical drawback is the memory complexity of the primer $\mathcal{F}$, which is $\mathcal{O}(N^2 \times d_{\text{model}})$ where $N$ is the number of channels. This scales quadratically increase with the number of variables and is a major limitation for high-dimensional datasets (e.g., Traffic N=862). The paper defers the solution (a GNN-based sparsification) entirely to Appendix D.2. This is a critical methodological detail for practical application and should be at least summarized in the main paper.


W2. The main paper (Sec 5) presents random initialization and lead-lag (FFT-based) initialization as options for $\mathcal{F}$. However, the ablation study in Appendix G.1 (Fig 3) strongly suggests that "Full" initialization (using lead-lag and other features) is significantly better than "Random." This dependency on pre-calculated features (similar to LIFT) somewhat undermines the "end-to-end" learning narrative and should be discussed more transparently.

**Questions:**

Q1. For the high-dimensional datasets (Traffic N=862, ECL N=321), was the full $\mathcal{O}(N^2)$ primer $\mathcal{F}$ instantiated, or was the GNN-based sparsification (Appendix D.2) used? If the sparse version was used, this is a critical experimental detail that should be in the main paper, as it changes the method being evaluated.

Q2. Could you please clarify the performance of prime attention with random initialization in the main results tables (Table 1 & 2)? Figure 3 (Appendix) suggests its performance is notably worse than the "Full" initialization. How much of the performance gain is attributable to the dynamic priming mechanism itself, versus the injection of pre-calculated FFT features at initialization?

Q3. The primer $\mathcal{F}_{i,j}$ is applied to both the Key (K) and Value (V) vectors. Did you ablate the impact of applying it only to K (modulating attention scores) versus only to V (modulating aggregated values)? Is the dual application necessary?

---

> ### Author Response · Authors · 2025-11-12
> **Response on weaknesses and questions**
>
> Thank you so much for your thorough review. We hope our response can clarify your concerns and we will work toward improving our paper per your recommendations.
>
> A1: The full primer was instantiated for all datasets, including Traffic N=862 and ECL N = 321. The sparsification strategy was never a necessity in our experiments. We will make this clear in the experiment section. Thank you for pointing this out.
>
> A2: The performance profile when using random vs. full initialization strategies for the primer was not clear or detailed enough in Figure 3 and we appreciate your thorough review of this. We actually find that the performance is mixed (with full initialization being better in most cases). On datasets like weather, full initialization performs around 1.5% better than random initialization. However, on exchange dataset, full initialization is around 0.6% WORSE than random initialization. We believe that specialized initialization can be beneficial when prior knowledge exists (such as lead-lag patterns in weather) but random initialization can actually be quite robust when clear patterns are either not known or non-existant. We do use full initialization in our experiments to keep things consistent. We will update Figure 3 and the corresponding section to better clarify the nuanced performance profiles of different primer initialization strategies and to roll back any and all over-claiming and over-simplification used in this section. However, we do not believe that having initialization strategies based on known patterns necessarily undermines the "end-to-end" learning narrative. Injecting known information as a learnable bias is an increasingly favored practice in deep learning and in our case it is completely optional (for example, weather shows 6.5% improvement with full initialization but random initialization still shows around 5% improvement over standard attention, which is a substantial improvement).
>
> A3: We haven't performed a systematic ablation of applying the primer separately to K and V because we found early on that having the primer applied to both K and V resulted in the most robust and stable performance. Intuitively, this makes sense because if the primer is not applied to both, then token i will essentially see two different versions of token j (one standard and the other primed), which may lead to performance deg or instability. Applying the primer only to K was by far the worst and applying it only to V was better but both were not as good as applying to both K and V.
>
> We will absolutely discuss the GNN sparsification strategy in more detail in the main paper, thank you for pointing this out.

---

> ### Author Response · Authors · 2025-11-17
> **Author response on memory complexity being a practical drawback**
>
> We make a separate comment to address prime attention's memory complexity issues. We agree with the reviewer that vanilla prime attention does introduce a significant drawback in increase in memory requirements. While this was not an issue in our experiments and we never resorted to sparsification strategies (even on the high dimensional datasets like traffic and ECL), we still agree that this can pose practical drawbacks as mentioned by the reviewer.
>
> We find that prime attention is potent enough to actually require less memory than standard attention while achieving comparable performance. With section G.2 and G.4 both showing, independently, that prime attention can outperform standard attention with lower memory usage by using less connectivity and less sequence length, we believe this provides a strong justification for prime attention's practical adoption. Below, we provide some additional results (extending section G.2), showing an instance of prime attention using around 400MB LESS memory compared to standard attention while achieving comparable performance.
>
> We provide prime attention's performance (in MSE) vs. memory usage on the Weather dataset below (we provide this as a table here for latex compatibility but will make it a chart in the updated paper for better visualization) at various channel connectivity densities.
>
> | Adj. Density   | Memory     | Performance |
> | -------------- | ---------- | ----------- |
> | 0.2            | 765.21 MB  | 0.180       |
> | 0.4            | 1212.75 MB | 0.179       |
> | 0.6            | 1691.39 MB | 0.177       |
> | 0.8            | 2177.59 MB | 0.176       |
> | 1.0 (full adj) | 2664.64 MB | 0.175       |
>
> For reference, standard fully-connected attention achieves 0.177 MSE using 2102.92 MB in this setup, requiring around 400 MB MORE than prime attention for comparable performance.
>
> ### Additional results on memory complexity
> We provide additional results to test the practical limits of prime attention below (using the fully instantiated primer).
>
> | channel count | memory (MB) |
> | ------------- | ----------- |
> | 200           | 910.97      |
> | 400           | 3533.47     |
> | 600           | 7887.04     |
> | 800           | 13990.02    |
> | 1000          | 21836.74    |
> | 1200          | 31433.40    |
> | 1400          | 42761.18    |
> | 1600          | 55836.52    |
> | 1800          | OOM         |
> | 2000          | OOM         |
> We create a synthetic dataset with variable channel-count filled with dummy data and record peak memory usage during training. The model is run on a single NVIDIA A100 80GB GPU using the same experimental settings as the Traffic dataset. We observe quadratic memory growth and OOM starting with receptive field of 1800 channels. We will include this as a chart in our ablations.
>
> While we certainly confirm that memory growth is an issue, we believe that being able to run nearly twice the channel-count of the largest benchmark dataset (Traffic) using the full primer (without resorting to sparsification strategy) shows practical utility. Overall, we would argue that the introduction of prime attention as a technique itself is the main contribution of our paper. While we start the conversation with our GNN-based sparsification strategy, we leave prime attention's adoption to truly massive datasets (with N > 1800) to future work.

---

### Author Response · Authors · 2025-11-29
**Special Note to Program/Area Chairs**

We sincerely thank the AC for their increased involvement in the review process this year amidst unprecedented circumstances. We provide a quick note here to voice our thoughts and to summarize the discussion thus far.

TL;DR
- Despite being borderline in score, ALL reviewers were positive on our novelty, contribution and the problem being solved.
- ALL reviewers viewed our empirical findings positively.
- Only one reviewer brought up novelty, but it was NOT questioning our novelty or its validity. Rather, it was a request to better position our novelty against other paradigms such as global gating methods, which we have done in our response.
- All other reviewer concerns were asking us to provide more details of our paper's setup, which we have done through addtional experiments and more descriptive explanations.
- Overall, all reviewers were genuinely interested in learning more about our paper, and we believe our paper has actually been strengthened as a result of addressing their insightful questions.

First, we strongly believe that our paper deserves a closer look by the AC because while we are borderline in score, **the overwhelming majority of reviewer concerns were on clarifications and additional experiments and NOT on lack of novelty or contribution of our paper**. We have since addressed all clarification requests and provided all requested additional experiments, which we believe that all reviewers would have been satisfied with. As the AC may agree, **our reviewers were genuinely interested in learning more about our paper and having their questions addressed**, which makes the cancellation of the discussion period all the more regrettable. We enjoyed very high quality reviews and positive reviewers and believe that our answers to their questions strengthen our paper.

Only one reviewer addressed our paper's novelty, but it was not questioning our paper's novelty, but rather a clarification request on our paper's positioning against other paradigms (such as agasinst global gating approaches), requesting for clarification and additional experiments, which we have done. All other reviewer concerns were regarding scalability of our proposed approach or questions on the details of our setup, both of which required us to run additional experiments and to make our ablation studies more descriptive and thorough. **We have since addressed all of these requests, which we believe actually strengthen our paper in many fronts**. For scalability concerns (which were the main concerns among the reviewers), we show with new experiments that we can handle around double the size of the biggest benchmark dataset used (Traffic dataset) in a single GPU and provide transparent analysis on exactly where our model starts failing. **Since most reviewers were concerned with whether we could handle Traffic dataset without compromise, showing that we can handle up to double it's size should adequately address their concerns**. For all other questions, we either perform new experiments (such as comparison against multiheaded attention with comparable parameter count) or make our existing analysis more descriptive.

---

### Meta-Review · Area_Chair_xNty · 2026-01-07

**Summary:**

The Reviewers recognized the clear motivation and positive empirical results of this paper. However, they also raised several key limitations and issues, including memory complexity bottlenecks and lack of validation for the sparsification strategy, insufficient comparisons with other advanced attention mechanism, vague experimental details and over-reliance on the appendix for core evidence. After the authors' rebuttal there are still issues rasied by reviewer that have not been resolved. Therefore, I suggest to reject the paper.

**Reviewer Concerns:**

Through the authors' further clarification and supplementary experimental results, several concerns have been addressed, including:
- Experimental details and controversies over initialization strategies (**Reviewer 1z1E**);
- F-matrix analysis and issues regarding paper structure (**Reviewer arqk**);
- Rigor of experimental statistics, and implementation details of FFT initialization (**Reviewer yrUp**).

However, several critical issues remain insufficiently addressed:
- Poor scalability in terms of memory overhead (**Reviewer 1z1E, yrUp**):
    Although the authors proposed a GNN-based sparsification strategy, this strategy has not undergone extensive validation, and it fails to address the fundamental issue of quadratic complexity growth.
- Inconsistent initialization strategy (**Reviewer 1z1E, yrUp**):
    The authors acknowledged that the initialization strategy relies on the biased characteristics of datasets, yet they failed to propose any potential adaptive adaptation strategy.
- Lack of empirical comparison with gating attention-based methods (**Reviewer arqk**):
    The authors only differentiated the two to a certain extent at the conceptual level, but did not provide further experimental results to demonstrate the advantages of the proposed method over simple gating attention-based approaches.

**Reviewer Scores:**

Given that the reviewers’ concerns have not been fully addressed and certain inherent methodological issues (e.g., memory overhead) remain, they seem unlikely to give a more positive score.

---

### Decision · Program_Chairs · 2026-01-26

Reject